# Network intrusion detection using a hybrid graph-based convolutional network and transformer architecture

Peter Appiahene[1*], Samuel Opoku Berchie[1*], Emmanuel Botchway [1*],
Michael Junior Ayitey[1], John Kwao Dawson[2], Henry Nii-Armah Mettle[1], Stephen Afrifa[1]

**1** Department of Information Technology and Decision Sciences, University of Energy and Natural Resources, Sunyani, Ghana, **2** Department of Computer Science, Sunyani Technical University, Sunyani, Ghana

\* peter.appiahene@uenr.edu.gh (PA); soberchie@gmail.com (SOB); emmanuel.botchway@uenr.edu.gh (EB)

## Abstract

Cloud computing continues to expand rapidly due to its ability to provide internet-hosted services, including servers, databases, and storage. However, this growth increases exposure to sophisticated intrusion attacks that can evade traditional security mechanisms such as firewalls. As a result, network intrusion detection systems (NIDS) enhanced with machine learning and deep learning have become increasingly important. Despite notable advancements, many AI-based intrusion detection models remain limited by their dependence on extensive, high-quality attack datasets and their insufficient capacity to capture complex, dynamic patterns in distributed cloud environments. This study presents a hybrid intrusion detection model that combines a graph convolutional layer and a transformer encoder layer to form deep neural network architecture. Using the CIC-IDS 2018 dataset, tabular network traffic data was transformed into computational graphs, enabling the model called "GConvTrans" to leverage both local structural information and global context through graph convolutional layers and multi-head self-attention mechanisms, respectively. Experimental evaluation shows that the proposed GConvTrans obtained 84.7%, 96.75% and 96.94% accuracy on the training, validation and testing set respectively. These findings demonstrate that combining graph learning techniques with standard deep learning methods can be robust for detecting complex network intrusion. Further research would explore other datasets, continue refining the proposed architecture and its hyperparameters. Another future research direction for this work is to analyze the architecture on other graph learning tasks such as link prediction.

**Data availability statement:** All relevant data are within the manuscript and its Supporting Information files.

**Funding:** The author(s) received no specific funding for this work.

**Competing interests:** The authors have declared that no competing interests exist.

## 1. Introduction

Cloud computing is defined as the delivery of hosted resources, including servers, databases, networking infrastructure, software, and data storage, via the internet [1]. Research demonstrates that, since its introduction, the global use of cloud-based applications and IT services has risen substantially and continues to grow [2]. The usage of these online services saves time, money, and storage space because almost all of the applications used are cloud-based [3]. Studies have shown that rapid application and service deployment made possible by cloud computing enables developers to quickly create resources and test new concepts [4]. A lot of companies and individuals are using Cloud computing due to these benefits, and more and more applications are moving to cloud settings. The security and privacy of the cloud resources and services that are being implemented are also gaining growing attention. This is because cloud services are being offered over the Internet. For instance, at the network layer with the architecture, cloud computing is vulnerable to a variety of traditional attacks, including IP spoofing, Address Resolution Protocol (ARP) spoofing, Routing Information Protocol (RIP) attack, DNS poisoning, man-in-the-middle attack, port scanning, insider attack, Denial of Service (DoS), Distributed Denial of Service (DDoS), and others. To deal with these types of attacks, Cloud service providers such as Amazon Elastic Compute Cloud, Microsoft Azure, Open Nebula, and others are utilizing Firewalls. Firewalls are regarded as the initial line of protection and safeguard a system's front-end access points. However, intrusion attacks are not detectable by firewalls since they only sniff network packets at the network perimeter. Furthermore, certain DoS or DDoS attacks are too complex for conventional firewalls to identify. Consequently, it is ineffective to use a typical firewall alone to prevent all breaches; additional security measures are needed.

In addition, a common method of attack prevention in cloud-based systems is the integration of a Network Intrusion Detection System (NIDS). Serving as an alarm system, NIDS improves security by detecting and marking network intrusions that successfully compromise the cloud-based system. In recent years, it has become common practice to build detection models for NIDS using machine learning and deep learning methods [5]. Even with the notable gains in efficiency over conventional methods, the majority of models still depend on a large amount of attack instance data. The network system of a company or enterprise produces lower quality attack sample data for training in real-world situations. Consequently, the detection capabilities of deep intrusion detection models based on this data are limited. In order to address this issue, research has demonstrated that Transformer and its variations, which make use of self-attention processes, have demonstrated notable success in completing Natural Language Processing (NLP) tasks like machine translation, conversation recognition, and text classification. It has been demonstrated that this approach is an effective way to identify network intrusions.

Transformer's main concept involves pre-training on a big corpus of text and then fine-tuning the learned model on a smaller dataset tailored to a particular task. Furthermore, several researchers have employed Transformer to identify intrusions

and abnormalities after learning about its capacity to manage ordered data sequences, demonstrating its resilience in a variety of situations. The majority of current models lack the capacity to learn and lose time series features because network intrusion is typically a continuous behavior over time. While certain Recurrent Neural Network (RNN)-based techniques are capable of learning time series features, their serial-based training approaches suffer from low convergence efficiency and lengthy training times. By efficiently learning the temporal correlation of network intrusion data, the Transformer's attention mechanism can improve the precision of network intrusion detection. Previous studies developed and put into practice this technique by utilizing the Transformer model's attention mechanism [6] in conjunction with network intrusion detection concepts. These previous studies benchmarked their proposed model against the CNNLSTM model and gradually increased the encoder layers for additional training. According to the final results, the Transformer-based network intrusion detection model demonstrated its effectiveness in anticipating network intrusions in cloud environments by achieving a prediction accuracy of over 93% under the given experimental conditions, which is comparable to the most recent approach on the CNN-LSTM model [7].

Despite achieving a prediction accuracy that outperformed earlier research, the Transformer-based network intrusion detection model is unable to recognize intricate infiltration patterns in dispersed environments like cloud datacenters. Initial research suggested the algorithm was not suitable for difficult environments like edge cloud systems. Due to their decentralized structure, these systems present particular difficulties that have not been addressed by earlier research. A common technique for building predictive models with graph-structured data is the use of graph neural networks (GNNs), which are mathematical models that can learn functions over graphs. As GNNs capture the reliance of graphs through message passing between graph nodes, they apply the predictive potential of deep learning to complex data structures that represent objects and their interactions. Because of their capacity to simulate intricate networks, Graph Neural Networks (GNNs) have become a potential method for network intrusion detection.

They have a distinct edge because of their innate capacity to process and evaluate graph data, particularly when working with intricate network patterns and structures. GNNs can represent network architecture as a graph and extract patterns from it, which makes it easier to analyze unstructured data in intrusion detection. Graph Neural Networks (GNNs), a type of deep learning model designed to assess data structured as graphs, where nodes represent entities and edges represent relationships between them, enable tasks such as node categorization, link prediction, and graph-level prediction. In IoT environments Graph Neural Networks, or GNNs, are effective tools for identifying intricate intrusion patterns. In order to improve our Transformer-based model's capacity to recognize intricate infiltration patterns in dispersed settings like cloud datacenters, the current study suggests integrating Graph Neural Networks into the Transformer-based model. Using the CIC-IDS 2018 dataset, the study further investigates the viability of employing Graph Neural integrated into the Transformer-based model for network intrusion detection and validates its prediction effect.

The study provides a hybrid intrusion detection model based the integration of Graph Neural Networks into Transformer that is optimized to recognize intricate infiltration patterns in dispersed settings like cloud datacenters.

The study provides a comprehensive literature on the proposed methods and concepts. In particular, we first discuss Graph Neural Networks followed by the Transformer model's architecture. The study also provides a detailed methodology used for the intrusion detection within cloud systems.

Finally, the study elaborates on the experimental setup, setting, and dataset while conducting a comprehensive evaluation of our methods using the CIC-IDS 2018 dataset and performance indicators. The various experimental results demonstrate our hybrid model resilience and strength.

The rest of the research paper is structured as: First, the study discussed the relevant literature that supports the problem statement and main objective. The study then presents the hybrid model, explains the algorithm used, and reveals the architecture of our created model. The results of the study are presented and discussed with relevant literature of, and our work is summarized in the conclusion.

## 2. Related works

Deep learning (DL) techniques have emerged as powerful tools in network intrusion detection systems (NIDS) due to their capacity to model complex and nonlinear patterns in high-dimensional traffic data. This section reviews existing studies that employ deep learning, with particular emphasis on graph-based approaches and transformer architectures [28,31]. Conventional NIDS techniques often struggle to capture relational dependencies among network entities and to model long-range temporal dynamics within network traffic. Graph neural networks (GNNs) address this limitation by effectively representing structural relationships, while transformers offer strong capabilities for sequence modeling and attention-based feature extraction. Nevertheless, the combined use of graph-based models and transformer architectures for intrusion detection remains largely underexplored in the current literature.

GNNs excel at encoding local topological patterns but struggle with scalability and noisy edges in large networks. For instance, Devnath [8] introduced GCNIDS, a GCN-based intrusion detection system tailored for Controller Area Networks (CAN) in vehicles, achieving superior accuracy and recall compared to traditional methods. Similarly, Pei et al. [2] proposed ResGCN, which incorporates attention-based deep residual modeling to address sparsity and nonlinearity in attributed networks, enhancing anomaly detection performance.

Early methods like E-GraphSAGE demonstrated that incorporating edge features (e.g., packet counts, connection types) alongside node attributes improved anomaly detection in IoT ecosystems but suffered from overfitting in heterogeneous environments [9,10].

To address noisy edges, Behavior Similarity Graph Attention Network (BS-GAT) introduced weighted attention based on node behavior rules, achieving 93% multi-class accuracy by refining neighbor aggregation [11]. Concurrently, Scattering Transform with E-GraphSAGE (STEG) combined multi-resolution edge analysis with graph convolutions, enabling detection of subtle traffic anomalies invisible to statistical methods [12]. Despite the success of these enhancements, GNNs suffer from some fundamental limitations such scalability issues, homogenized node embeddings in deep GNNs and non-informative connections among nodes [13]. Recent solutions like Performance-Adaptive Sampling Strategy (PASS) mitigate these issues by jointly training a learnable sampler and GNN. PASS dynamically selects task-relevant neighbors using importance scores derived from node embeddings, reducing computational costs by 60% while boosting accuracy on LinkedIn job recommendation graphs [13]. Similarly, Adaptive Feature and Topology GCN (AAGCN) fused multi-scale feature convolutions with topology-aware embeddings to handle feature sparsity in citation networks, improving Cora dataset accuracy by 7.2% over vanilla GCN [10].

Hybrid models combining GCNs with attention mechanisms have also been explored. Jahin et al. [14] developed CAGN-GAT Fusion, merging Contrastive Attentive Graph Networks with Graph Attention Networks, demonstrating robust performance across multiple benchmark datasets, including KDD-CUP-1999 and CICIDS2017. Transformer architectures have been leveraged to capture long-range dependencies in network traffic. Pure transformer-based NIDS models like E-T-GraphSAGE (ETG) treat traffic flows as temporal token sequences, achieving 93% accuracy on CIC-IDS 2018 by learning protocol-specific signatures without manual feature engineering [15]. Farrukh et al. [16] introduced XG-NID, a dual-modality framework combining heterogeneous GNNs with large language models to analyze flow-level and packet-level data, achieving an F1 score of 97% in multi-class classification. The fusion of CNNs and transformers has been further explored to enhance feature extraction. The ImagTrans model [17] employs CNN-based position encoding and ConvTrans blocks to extract local and global features from IoT network traffic, improving generalization and convergence speed. Furthermore, Ullah et al. [18] presented IDS-INT, a transformer-based transfer learning method combined with a CNN-LSTM model to detect various attack types in imbalanced datasets. Moreover, a survey [19] discussed the role of deep learning in proactive cybersecurity measures, emphasizing the potential of transformer-based models like TGAN-AD in anomaly detection across various datasets.

Beyond cybersecurity, Relational Graph Transformers address relational database complexities through multi-element tokenization (node features, type, hop distance), outperforming GNNs by 18% on RelBench benchmarks by modeling

cross-table dependencies [9]. In the realm of remote sensing, Ali et al. [20] proposed N-STGAT, a spatio-temporal graph neural network-based intrusion detection system, integrating Graph Attention Networks (GAT) with Long Short-Term Memory (LSTM) networks to enhance detection reliability.

## 3. Materials and methods

This section outlines the experimental approach adopted in this study. It incorporates every stage in the workflow; from data collection to data modelling and evaluation, into a conceptual framework. Each step is carefully and thoroughly examined in the following subsections.

### 3.1. Study workflow

The framework consists of two pipelines, the data preprocessing and the model training pipeline. In the preprocessing pipeline, the various data files for each date in the dataset are loaded up in chunks and processed. The data which is retrieved in tabular form is converted to a graph. The most significant features were retained while the rest (which were mostly statistical) were excluded to make subsequent analysis and processing easier and manageable. The data was then split into training, validation and test subsets. The categorical features are encoded and numerical features scaled. We apply SMOTE to oversample the minority classes as a mean to minimize skew towards the majority class. We the group the various rows into clusters based on the time window and class in which they fall. Edges are then formed between each node (row or entry) and its closest 50 neighbors. Edges are deduplicated to ensure the robustness of the output graph (Fig 1).

In the model training pipeline, the proposed architecture and the baseline models are trained and evaluated on the graph data. First the graph is loaded is neighbors are sampled via adjacency lists in order to create dataloaders for the training, validation and test subsets. The models are trained and evaluated in an iterative manner. The node embeddings for the model's predictions on a section of the test set is visualized (Fig 2).

### 3.2. Data collection

The study used the publicly accessible CSE-CIC-IDS2018 dataset, created by the University of New Brunswick for analyzing distributed Denial-Of-Service (DDoS) data and can be found at https://www.unb.ca/cic/datasets/ids-2018.html. The data consists of several server logs of DDoS data over a period in 2018 which have been saved as comma separated value (csv) files for each specific date of logging. The dates on which the data was logged are as follows; 14th, 15th, 16th, 20th, 21st, 22nd, 23rd, 28th of February and the 1st and 2nd of March. Each file of the dataset contains eighty columns, each of which represent an entry in the Intrusion Detection System (IDS) logging system used by the data curator. The data curator's IDS categorizes network traffic in two directions, forward and backward, and columns are assigned for values associated with direction. The dataset consists of 80 different features and over one million rows with 14 different classes. We focus on the 14th February data in this study.

### 3.3. Data preprocessing

Data preprocessing is an important part of any machine learning pipeline as it prepares the input data in a format that is suitable for training machine learning or deep learning models [21]. Effective preprocessing can improve model performance by making the input features more normalized and representative of the inherent patterns within the data [22].

Only the most relevant attributes were retained from the original dataset to reduce noise and computational overhead. Specifically, the fields 'Dst Port', 'Protocol', 'Timestamp', 'Flow Duration', 'Tot Fwd Pkts', 'Tot Bwd Pkts', and 'Label' were extracted. The 'Protocol' column was cast to a categorical data type to facilitate later one-hot encoding of categorical protocols, while 'Flow Duration' was converted to a numeric type. Any records with negative flow durations (indicating

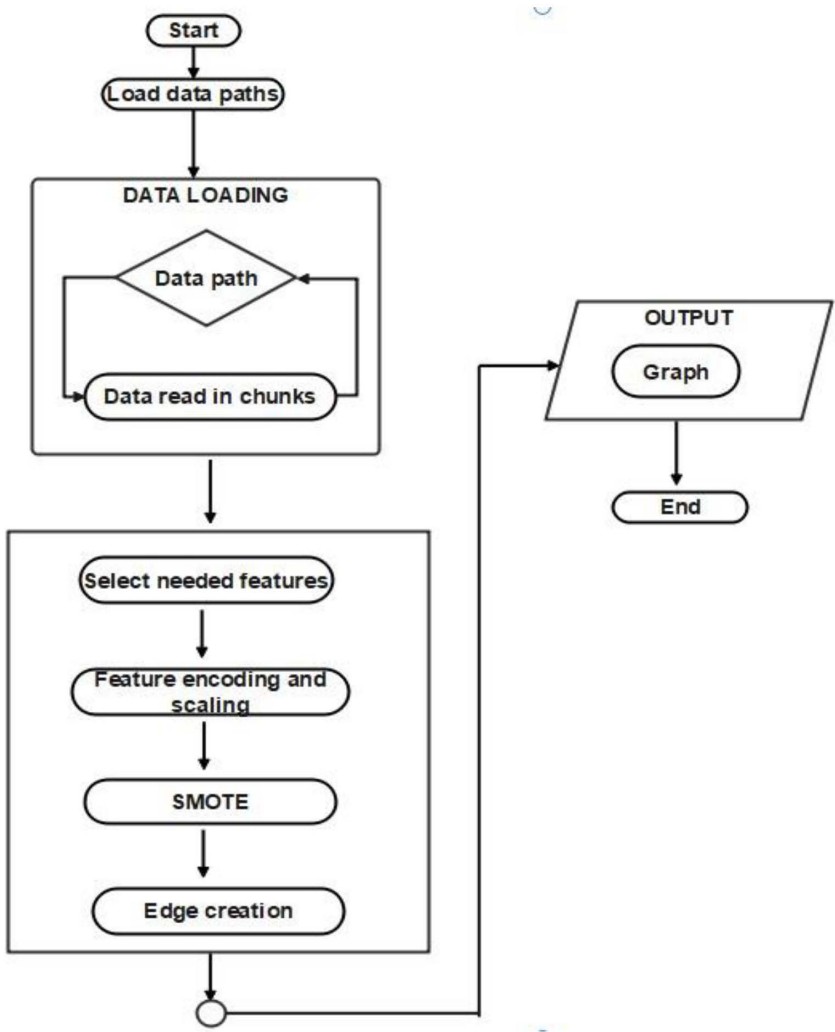

**Fig 1. A flowchart of the data preprocessing pipeline.**

erroneous measurements) were discarded to prevent skewed transformations. After filtering, the index was reset to maintain data integrity.

To stabilize variance and reduce the impact of extreme values, a log-transform was applied to the cleaned flow durations. Each duration (in microseconds) was divided by $10^6$ and then transformed via $log_{1p}(\cdot)$. This compresses the range of values, making the distribution more amenable to learning. Subsequently, flows were categorized into four bins, 'short' ($0 < $log duration$ \leq 1$), 'medium' ($1 < $log duration$ \leq 10$), 'long' ($10 < $log duration$ \leq 60$), and 'very_long' (log duration$ > 60$). Temporal locality is crucial for constructing meaningful edges in the graph. Each timestamp was floored to the nearest hour ("Time Window"), grouping flows into one-hour intervals. This discretization facilitates the formation of time-aware groupings, ensuring that only flows occurring within the same hour can be connected. The integer representation of the floored timestamp (seconds since epoch) was stored temporarily for downstream computations, then replaced with the original datetime once grouping indices were established. Port numbers exhibit different semantic meanings based on their range: well-known (0–1023), registered (1024–49151), and ephemeral (49152–65535). The destination port ('Dst Port') was converted to an integer type and binned into a new feature 'Port Category'. This categorical feature captures the service

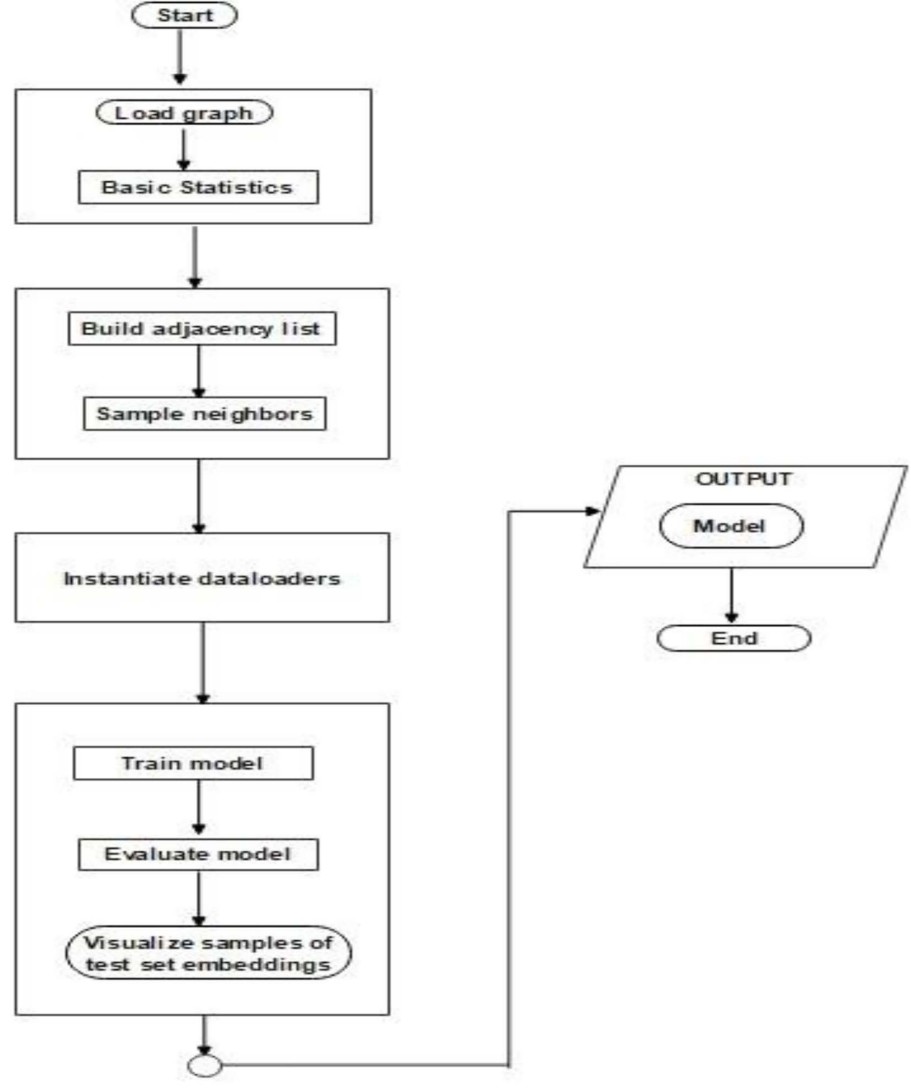

**Fig 2. A flowchart of the model training pipeline.**

class of each flow. To prevent data leakage, the dataset was randomly shuffled and partitioned into 60% training, 25% validation, and 15% test splits. A split map array was maintained to ensure split-aware processing during grouping and edge construction. The numerical features 'Flow Duration', 'Tot Fwd Pkts', and 'Tot Bwd Pkts' were scaled exclusively on the training subset, then applied to all records. This z-score normalization centers each numeric column to zero mean and unit variance to improve convergence during graph learning. The categorical features Protocol, Duration Bin, and Port Category were one-hot encoded. Network traffic datasets often exhibit significant class imbalance. To mitigate this, Multiclass SMOTE was applied to the training feature set of the data. Synthetic Minority Oversampling Technique (SMOTE) generates new samples for underrepresented classes by interpolating between existing minority-class instances [23,24]. After oversampling, the training set was truncated or padded back to its original size (60% of total records) to maintain consistent batch dimensions. The oversampled labels were reconverted to one-hot form. This procedure preserves class proportions during evaluation while providing the graph-building stage with a balanced training set.

### 3.4. Proposed architecture

Building on Kipf et. al's work [25], we propose a hybrid architecture, Graph Convolution Transformer Network (GConvTrans), which combines graph convolutions with a transformer encoder. First, we describe the two main components of the architecture, the graph convolution layer and the transformer encoder layer. We then detail the model architecture which combines the two components and the classification head.

**3.4.1. Graph convolution layer.** The first stage of the proposed architecture applies a graph convolution layer to the input node features. Given a feature matrix $X \in R^{N \times d_{in}}$ and an edge index indicating connectivity, the convolution operation aggregates information from each node's immediate neighbors. The operation starts with the normalization of the adjacency matrix of the input graph to ensure numerical stability. The normalized adjacency matrix is obtained by:

$$\widetilde{A} = D^{-\frac{1}{2}} \cdot AD^{-\frac{1}{2}} \tag{1}$$

Where $D$ is the diagonal degree matrix of $A$. The normalized adjacency $\widetilde{A}$ then propagates node features through a linear transformation and nonlinear activation. Mathematically,

$$H^{l+1} = \sigma \left( \widetilde{A} H^{(l)} W^{(l)} \right) \tag{2}$$

Where $W \in R^{d_{in} \times d_h}$ is a trainable weight matrix, and $\sigma$ denotes a nonlinear activation function which is ReLU in this case. ReLU introduces non-linearity and helps to handle vanishing gradients. This layer encodes the local structural patterns within the graph and node attributes into a hidden representation embeddings of dimension $F_{hid}$.

**3.4.2. Transformer encoder layer.** $H^l$ is passed through a stack of $L$ TransformerEncoder layers. Each layer employs multi-head self-attention (with $h$ heads) and a feed-forward network with GELU activation, wrapped in pre-norm residual connections. By treating the $N$ node embeddings as a sequence, the Transformer can learn to reweight and combine information from distant nodes.

**3.4.3. Classification head.** After the transformer encoder layer, we apply dropout to the refined embeddings and project each node's vector into the label space of size $C$ via a linear layer. A log-softmax activation produces final class probabilities for each node (Fig 3):

$$\hat{y} = \log \left( \frac{exp\left(x_i\right)}{\sum_j exp\left(x_j\right)} \right) \tag{3}$$

### 3.5. Performance evaluation metrics

To evaluate the performance of the proposed architecture, the cross-entropy loss and accuracy metrics are used across all the data splits. All metrics are calculated only on the nodes specified by the corresponding mask in the graph data.

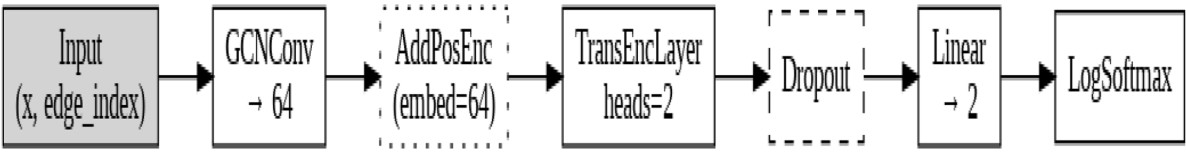

**Fig 3. The proposed architecture.**

**3.5.1. Cross-entropy loss.** The primary training objective is the average negative log-likelihood of the correct class over a set of nodes S. Given model outputs $\hat{y} \in R^{N \times C}$, where $\hat{y}_{i, c}$ is the predicted probability that node $i$ belongs to class c, and one-hot labels $\hat{y}_{i, c} \in \{0, 1\}$, the loss can be obtained by:

$$Loss = -\frac{1}{|S|} \sum_{i \in S} \sum_{c=1}^{C} y_{i,c} \log(\hat{y}_{i, c})$$

(4)

**3.5.2. Accuracy.** Accuracy measures the number of correctly classified or predicted nodes. The accuracy can be obtained by:

$$Accuracy = \frac{\sum (TP, \ TN)}{\sum (TP, \ TN, \ FP, \ FN)}$$

(5)

## 4. Results

This section presents the configurations and results obtained in the processes observed in the methodology of this study.

### 4.1. Experimental set-up and configurations

The entire experiment was set up and carried out on the Google Colaboratory (Colab) platform in Python software version 3.11.13 (The Python Software Foundation (PSF), 1209 Orange Street, Wilmington, DE, USA). The memory and disk space provided by the Colab environment were 12.0 gigabytes and 108 gigabytes respectively. Table 1 shows the system specifications and setup that were used in the study.

### 4.2. Experimental results

Using the graph creation approach earlier described, we are able to convert the tabular data into a computational graph. The graph consists of 14 different node features, 3 classes, 70554 nodes and 831794 undirected edges. Of the total number of edges, 19235 edges are isolated while 51319 edges are connected. The masking of nodes for the creation of training, validation and test subsets in the graph. These subsets consist of 42332, 17638 and 10584 nodes respectively (Fig 4).

The model is trained with the following hyperparameters: number of epochs of 30, 0.001 learning rate, batch size of 70. Other hyperparameters include 32 hidden channels, 4 heads in the multi-head attention components which are all parameters of the one-layer TransformerEncoder. Over the course of training, we noticed that the model showed signs of early convergence. Within the first five epochs, training loss falls sharply from around 1.0 to 0.46, and validation loss from 0.93 to 0.33, a corresponding jump in validation accuracy from 57% to 86%. Beyond epoch 5 the curves begin to plateau more gradually: by epoch 15 training loss has reached approximately 0.33 with training accuracy around 84.7% and validation

**Table 1. System specifications and setup used in the study.**

| Product | Specifications |
|---|---|
| CPU (Colab) | Intel(R) Xeon(R) CPU @ 2.00GHz |
| GPU (Colab) | Tesla T4 |
| Memory | 12 Gigabytes |
| Computer system type | 64-bit OS |

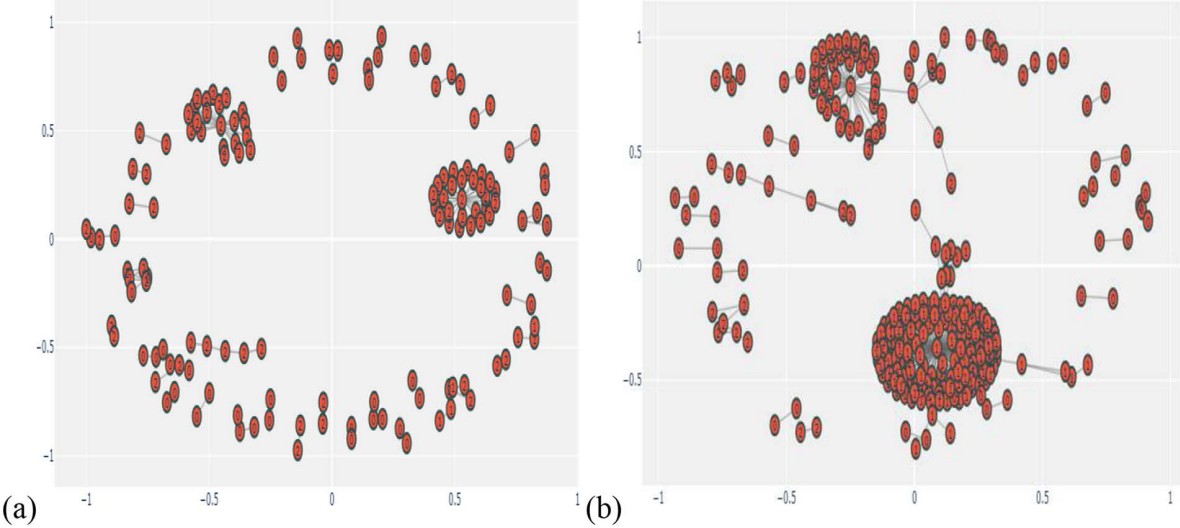

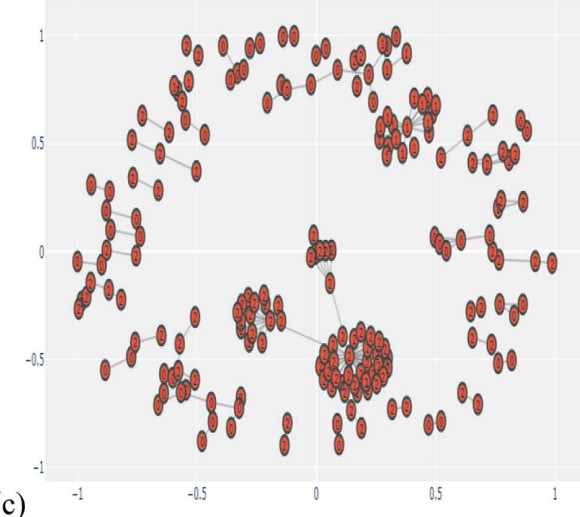

**Fig 4. Samples of nodes in the training, validation and test sets of the graphs.**

loss has stabilized around 0.17 with validation accuracy around 95.9%. Over the final ten epochs, we observe only small improvements, with validation accuracy peaking at 96.8% around epoch 24 and settling at 96.75% by epoch 29. On the held-out test set, GConvTrans achieved a final loss of 0.1488 and accuracy of 96.94% (Fig 5).

To probe the model's ability to identify the different node classes in the graph, we extract the 128-dimensional node embeddings produced by the final Transformer encoder and reduced them to two dimensions using t-SNE. Fig 6 shows 100 randomly sampled test-set nodes colored by class label. We observe 3 well-isolated groups or clusters each corresponding to one of three class labels 0, 1 and 2. Class 0 refers to the "Benign", Class 1 refers "FTP-BruteForce" and "SSH-Bruteforce". Class 0 forms the top-right cluster, Class 1 the lower-left, and Class 2 the mid-left. Even though the clusters are distinct and quite spread out, there appears to be some overlap in the mid-section.

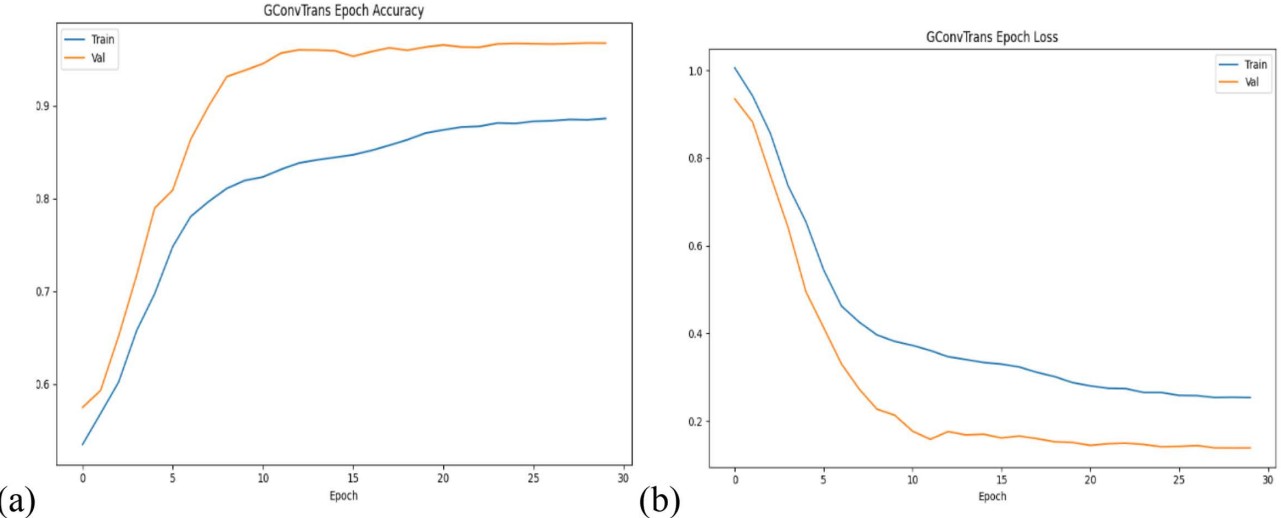

(a)  (b)

**Fig 5. The change in performance for both training and validation sets as training progresses.**

## 5. Discussion

The performance of the proposed model can be attributed to the structure of the architecture which is a hybrid network of the graph convolution and transformer encoders. The graph convolutional layers model the local structural relations through neighborhood aggregation. The transformer encoder layers integrate the global context from distant nodes using multiple self-attention heads. In the processing of the transformer layer, a chunking mechanism is applied to efficiently minimize memory usage and scalability issues.

To assess the model's performance, we compare our proposed architecture against state-of-the-art (SOTA) approaches in network intrusion systems. The proposed model achieves state-of-the-art performance on the CSE-CIC-IDS2018 dataset. Most of the existing studies built traditional deep learning on network intrusion tabular data rather than computational graph format. Our proposed model detects network intrusion via a graph representation of the data (Table 2).

While the aforementioned SOTA studies have demonstrated impressive results on various datasets, including NSL KDD used by Xi et al. [27], and Wang et al. [11], it is worth noting that their approaches predominantly rely on traditional deep learning architectures or transformer-based models. These methods, while achieving high accuracy, often treat network intrusion data as tabular or sequential inputs, potentially overlooking the intricate relational dependencies inherent in network traffic. For instance, Xi et al. [27] utilize transformer-based architectures that excel in capturing sequential patterns but do not explicitly model the graph-like structure of network interactions. This limitation may hinder their ability to fully represent the complex connectivity and dependencies between network entities, which are critical for detecting sophisticated intrusion patterns.

In contrast, our proposed model employs a graph learning approach, representing the data as a computational graph. This allows the model to capture both local structural relations and global contextual information through graph convolutions and transformer encoders, respectively. By leveraging the inherent graph structure of network data, our approach provides a more holistic understanding of the relationships between entities, offering a distinct advantage over the SOTA methods that do not adopt this perspective.

Moreover, while Farrukh et al. [16] and Wang et al. [11] incorporate heterogeneous graph neural networks (GNNs) alongside large language models (LLMs), their use of GNNs is supplementary, focusing on integrating textual data with graph representations rather than relying solely on the graph structure for intrusion detection. Our model, however,

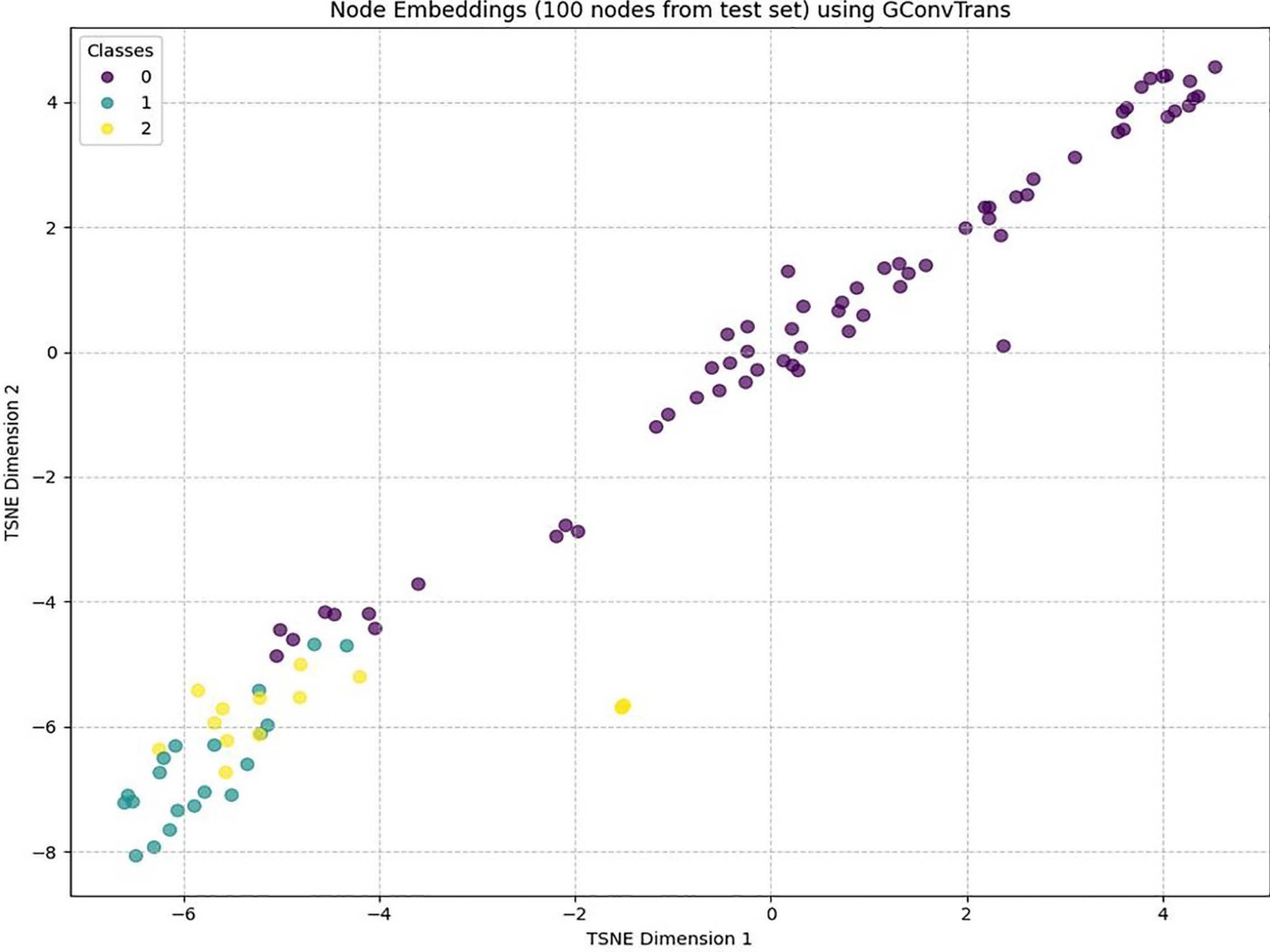

**Fig 6. Node embeddings from the proposed model on the test set.**

**Table 2. Comparing GConvTrans and recent state-of-the-art intrusion detection models.**

| Study | Year | Architecture | Dataset | Metrics |
|---|---|---|---|---|
| Current study | --- | Proposed architecture | CSE-CIC-IDS2018 | Training = 84.7%, Validation = 96.75%, Testing = 96.94% |
| Sharmin et al. [26] | 2023 | Deep Contractive Autoencoder (DCAE) | NSL-KDD; CIC-DDoS2017; CIC-DDoS2019 | Accuracy ranging from 93.41% to 97.58% |
| Xi et al. [27] | 2024 | Multi-scale Transformer (IDS-MTran) | NSL-KDD; CIC-DDoS2019; UNSW-NB15 | Accuracy: 99% |
| Wang et al. [11] | 2023 | Heterogeneous GNN + LLM (XG-NID) | NSL-KDD; CIC-IDS2017 | Accuracy: ~95% to 97% |

demonstrates that a pure graph learning approach—without the need for additional modalities—can achieve competitive performance. This suggests that the graph representation alone is sufficiently powerful for modeling network intrusion data, challenging the complexity introduced by hybrid GNN-LLM frameworks in these SOTA works.

Additionally, the proposed architecture achieves SOTA performance with a relatively shallow network, indicating that the combination of graph convolution and transformer encoders is highly efficient in extracting relevant features from graph-structured data. This efficiency could prove particularly valuable in resource-constrained environments, where the computational overhead of deeper transformer-based models or hybrid GNN-LLM systems might be prohibitive.

### 5.1. Limitation of the Study

One limitation of this study is that due to computation and storage shortfalls, the study only analyzed one dataset, which is the the CIC-IDS 2018 (Feb 14 subset). Consequently, the generalizability of the proposed model to other network traffic scenarios or intrusion datasets may be limited. Future work should evaluate the model on multiple datasets with diverse attack types and traffic patterns to validate its robustness and ensure broader applicability across different network environments.

Another limitation is the absence of detailed baseline and ablation comparisons due to compute constraints. Planned experiments including GCN only, Transformer only, MLP, XGBoost, and hybrid variants with individual components removed are still pending. These evaluations will be addressed in future work to clearly show the added benefits provided by the hybrid model.

## 6. Conclusion

This study presents a comprehensive framework for a hybrid deep learning architecture, combining a graph neural network and a transformer encoder layer, for network intrusion detection systems (NIDS) in IoT devices. A scientific experiment and evaluation of the proposed model on an IoT network log dataset have produced promising results and insights that can help in the development of effective network intrusion detection systems [29].

The experimental results reveal that the proposed architecture, GConvTrans, can achieve state-of-the-art results. GConvTrans obtained 84.7%, 96.75% and 96.94% accuracy on the training, validation and testing set respectively. Observations from the node embeddings of the proposed model show that the proposed architecture has the potential to correctly group nodes of the same class very well.

These findings add to the growing literature on the application of graph learning to cybersecurity and network intrusion detection through empirical, reproducible evidence which is uncommon in the existing studies. We emphasize that architectural, computational and training stability as well as efficiency are equally vital considerations in graph-based approaches in IoT and cloud security [7].

The proposed model demonstrates practical significance by offering efficient intrusion detection with lower computational and memory requirements compared to large transformer-based architectures. This makes it particularly suitable for deployment in real-world cloud and IoT environments, where resources are often constrained. By balancing detection performance with efficiency, the approach enables timely and scalable network security monitoring, supporting broader adoption in operational settings [30].

Future work will focus on evaluating the proposed model on multiple datasets with diverse attack types and network traffic patterns to enhance generalizability. Additionally, we plan to explore optimizations for real-time deployment in large-scale cloud and IoT environments, including lightweight model architectures and incremental learning approaches. Investigating the integration of graph-based and transformer techniques may further improve detection accuracy while maintaining computational efficiency.

## Acknowledgments

The authors give thanks to God for the strength, wisdom, and guidance that made this research possible.

## Author contributions

**Conceptualization:** Peter Appiahene, Emmanuel Botchway, Henry Nii-Armah Mettle.

**Data curation:** Peter Appiahene, Samuel Opoku Berchie, Emmanuel Botchway, Michael Junior Ayitey, John Kwao Dawson, Henry Nii-Armah Mettle, Stephen Afrifa.

**Formal analysis:** Peter Appiahene, Samuel Opoku Berchie, Emmanuel Botchway, Michael Junior Ayitey, John Kwao Dawson, Henry Nii-Armah Mettle, Stephen Afrifa.

**Investigation:** Peter Appiahene, Emmanuel Botchway, Michael Junior Ayitey, John Kwao Dawson, Henry Nii-Armah Mettle.

**Methodology:** Samuel Opoku Berchie, Emmanuel Botchway, Michael Junior Ayitey, Henry Nii-Armah Mettle, Stephen Afrifa.

**Project administration:** Peter Appiahene, Emmanuel Botchway, Michael Junior Ayitey.

**Resources:** Samuel Opoku Berchie, Emmanuel Botchway, Michael Junior Ayitey, Henry Nii-Armah Mettle, Stephen Afrifa.

**Software:** Samuel Opoku Berchie, Emmanuel Botchway, Michael Junior Ayitey.

**Supervision:** Peter Appiahene, Emmanuel Botchway, Michael Junior Ayitey.

**Validation:** Peter Appiahene, Emmanuel Botchway, Michael Junior Ayitey, John Kwao Dawson.

**Visualization:** Peter Appiahene, Samuel Opoku Berchie, Emmanuel Botchway, Henry Nii-Armah Mettle, Stephen Afrifa.

**Writing – original draft:** Samuel Opoku Berchie, Emmanuel Botchway.

**Writing – review & editing:** Peter Appiahene, Emmanuel Botchway, Michael Junior Ayitey, John Kwao Dawson.

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
