## [Decision Letter · Decision Letter 0]

17 Oct 2025

Dear Dr. Botchway,

Thank you for submitting your manuscript to PLOS ONE. After careful consideration, we feel that it has merit but does not fully meet PLOS ONE’s publication criteria as it currently stands. Therefore, we invite you to submit a revised version of the manuscript that addresses the points raised during the review process.

We look forward to receiving your revised manuscript.

Kind regards,

Burak Tasci, Ph.D.

Academic Editor

PLOS ONE

**Journal Requirements:**

1. When submitting your revision, we need you to address these additional requirements. Please ensure that your manuscript meets PLOS ONE's style requirements, including those for file naming. The PLOS ONE style templates can be found at https://journals.plos.org/plosone/s/file?id=wjVg/PLOSOne_formatting_sample_main_body.pdf and https://journals.plos.org/plosone/s/file?id=ba62/PLOSOne_formatting_sample_title_authors_affiliations.pdf 2. Please note that PLOS One has specific guidelines on code sharing for submissions in which author-generated code underpins the findings in the manuscript. In these cases, we expect all author-generated code to be made available without restrictions upon publication of the work. Please review our guidelines at https://journals.plos.org/plosone/s/materials-and-software-sharing#loc-sharing-code and ensure that your code is shared in a way that follows best practice and facilitates reproducibility and reuse. 3. We note that your Data Availability Statement is currently as follows: All relevant data are within the manuscript and its Supporting Information files. Please confirm at this time whether or not your submission contains all raw data required to replicate the results of your study. Authors must share the “minimal data set” for their submission. PLOS defines the minimal data set to consist of the data required to replicate all study findings reported in the article, as well as related metadata and methods (https://journals.plos.org/plosone/s/data-availability#loc-minimal-data-set-definition).  For example, authors should submit the following data: - The values behind the means, standard deviations and other measures reported;- The values used to build graphs;- The points extracted from images for analysis. Authors do not need to submit their entire data set if only a portion of the data was used in the reported study. If your submission does not contain these data, please either upload them as Supporting Information files or deposit them to a stable, public repository and provide us with the relevant URLs, DOIs, or accession numbers. For a list of recommended repositories, please see https://journals.plos.org/plosone/s/recommended-repositories. If there are ethical or legal restrictions on sharing a de-identified data set, please explain them in detail (e.g., data contain potentially sensitive information, data are owned by a third-party organization, etc.) and who has imposed them (e.g., an ethics committee). Please also provide contact information for a data access committee, ethics committee, or other institutional body to which data requests may be sent. If data are owned by a third party, please indicate how others may request data access. ?> 4. If the reviewer comments include a recommendation to cite specific previously published works, please review and evaluate these publications to determine whether they are relevant and should be cited. There is no requirement to cite these works unless the editor has indicated otherwise.

**Additional Editor Comments:**

Thank you for submitting your manuscript to PLOS ONE. Following editorial and peer review, several revisions are required to ensure that your work meets the journal’s standards for transparency, scientific rigor, and clarity. First, in alignment with PLOS ONE’s commitment to open science and reproducibility, we kindly request that you make the code used in your study publicly available via a trusted platform such as GitHub, GitLab, or Zenodo. Please include the corresponding access link in the Data Availability Statement of your manuscript.

Additionally, we recommend a thorough revision of the manuscript to improve the quality of the English language. This includes addressing grammatical errors, improving sentence structure, and ensuring overall clarity. You may consider using a professional editing service to ensure the language meets academic publishing standards.

We also encourage you to update your literature review by incorporating recent and relevant studies from the past few years. This will help contextualize your work within current research trends and reinforce the relevance of your findings.

Lastly, please prepare a detailed response letter addressing each of the reviewers' comments. For every point raised, clearly explain how it was addressed and indicate where the corresponding changes were made in the manuscript. If there are suggestions you choose not to implement, please provide a reasoned explanation. You are not obligated to include or cite any references suggested by reviewers if they are not directly relevant to your study. We encourage you to critically evaluate each suggestion and include only those that add value to your work.

We appreciate your efforts in improving the manuscript and look forward to receiving your revised submission.

Reviewers' comments:

**Comments to the Author**

1. Is the manuscript technically sound, and do the data support the conclusions?

Reviewer #1: Yes

Reviewer #2: Partly

2. Has the statistical analysis been performed appropriately and rigorously?

Reviewer #1: Yes

Reviewer #2: No

3. Have the authors made all data underlying the findings in their manuscript fully available?

Reviewer #1: No

Reviewer #2: No

4. Is the manuscript presented in an intelligible fashion and written in standard English?

Reviewer #1: Yes

Reviewer #2: Yes

**Reviewer #1:** Authors paper on “Network Intrusion Detection Using a Hybrid Graph-based Convolutional Network and Transformer Architecture” is a relevant topic. In this novelty and methodology is clear along with gap analysis. However, a few points can be improved before Submission as given below:

1. Consistency in results: Ensure that performance metrics (training/validation/test accuracies) are consistently reported across Abstract, Results, and Conclusion.

2. Broader dataset validation: Since only one dataset (CIC-IDS 2018, Feb 14 subset) was used, the generalizability claim is limited. PLOS ONE usually requires broader evidence or at least a stronger discussion of limitations.

3. Data availability: Already compliant (dataset is public), but consider including the exact preprocessing code repository for transparency.

4. Writing quality: Some grammatical issues and repetitive phrasing in Introduction/Related Works could be polished for readability.

5. Impact framing: Strengthen the claim of practical significance (e.g., efficiency in real-world cloud/IoT deployments, lower resource requirements compared to deep transformers).

**Reviewer #2:**  1-Grouping rows by time window and class is followed by connecting each node to its 50 closest neighbors. The input graph is constructed using only time window information and k-NN in feature space, without utilizing true labels.

2-Avoid performing random shuffles. Conduct chronological (or day-wise) partitioning on the CIC-IDS2018 dataset (for example, training on earlier days and testing on a later day). Generate distinct graphs for each partition (ensuring no edges cross between splits), or employ mini-batches that conceal message passing to prevent the GNN from observing validation/test nodes during training. Provide results as the mean ±95% confidence interval using a minimum of 5 different seeds.

3-Provide the precise measurement method (Euclidean or Cosine), feature standardization utilized in k-NN, the examined values of k, clarity on directed or undirected edges, and methodology for adjacency normalization. Include experimental variations: k in {10, 25, 50, 100}, with or without time-window restriction.

4-Compare the performance of models using GCN-only, Transformer-only, MLP, and a strong tabular baseline (e.g., XGBoost) on the same dataset split. Additionally, conduct component ablations by removing the GCN and Transformer components to assess the added value of the hybrid model over simpler ones. This analysis validates that the hybrid model offers tangible benefits beyond basic models.

5-Provide detailed performance metrics including per-class precision, recall, F1 score (macro and weighted), PR-AUC, ROC-AUC, confusion matrices, and calibration (ECE) on the test set. Present the test accuracy as a single value consistent throughout the report.

6-Provide a repository containing graph-building scripts, split masks, configurations, code, and random seeds. Include a list of package versions and hardware specifications. Clearly document hyperparameter search details, specifying the ranges, patience, and conditions for early stopping. Add descriptive captions to figures and diagrams that accurately reflect the pipeline implementation.

**Do you want your identity to be public for this peer review?** For information about this choice, including consent withdrawal, please see our Privacy Policy

Reviewer #1: No

Reviewer #2: **Yes:** Amjed Abbas Ahmed

---

## [Author Response · Author response to Decision Letter 1]

27 Nov 2025

RESPONSE TO REVIEWERS – REVISED MANUSCRIPT SUBMISSION

Manuscript Title: Network Intrusion Detection Using a Hybrid Graph-based Convolutional Network and Transformer Architecture

Manuscript ID: PONE-D-25-45283

We are grateful for the opportunity to revise and resubmit our manuscript titled “Network Intrusion Detection Using a Hybrid Graph-based Convolutional Network and Transformer Architecture”. We sincerely appreciate the insightful and constructive comments provided by the reviewers and the academic editor. Their feedback has significantly improved the clarity, quality, and overall contribution of our work.

We have carefully addressed each comment in detail and revised the manuscript accordingly. A comprehensive table summarizing the reviewers’ comments alongside our corresponding responses is presented in the Word Document titled "Response to Reviewers". All changes made to the manuscript have been clearly incorporated in the revised version, and page/section numbers have been indicated where appropriate.

We believe that the revisions we have made substantially strengthen the manuscript, and we hope the updated version meets the expectations of the reviewers and the academic editor. We appreciate your time and consideration, and we look forward to your favorable response.

Thank you once again.

Sincerely,

Emmanuel Botchway

---

## [Decision Letter · Decision Letter 1]

30 Dec 2025

Network Intrusion Detection Using a Hybrid Graph-based Convolutional Network and Transformer Architecture

PONE-D-25-45283R1

Dear Dr. Botchway,

We’re pleased to inform you that your manuscript has been judged scientifically suitable for publication and will be formally accepted for publication once it meets all outstanding technical requirements.

Kind regards,

Burak Tasci, Ph.D.

Academic Editor

PLOS One

Additional Editor Comments (optional):

Your submitted manuscript has been reviewed and accepted for publication in our journal. We congratulate you on this successful work and thank you for your contributions.

Reviewers' comments:

Reviewer's Responses to Questions

**Comments to the Author**

Reviewer #1: All comments have been addressed

2. Is the manuscript technically sound, and do the data support the conclusions?

Reviewer #1: Yes

3. Has the statistical analysis been performed appropriately and rigorously?

Reviewer #1: Yes

4. Have the authors made all data underlying the findings in their manuscript fully available?

Reviewer #1: Yes

5. Is the manuscript presented in an intelligible fashion and written in standard English?

Reviewer #1: Yes

Reviewer #1: Authors presented a work on "Network Intrusion Detection Using a Hybrid Graph-based Convolutional

Network and Transformer Architecture".

They worked on all review points and modified the paper.

At this stage it seems acceptable.

**Do you want your identity to be public for this peer review?** For information about this choice, including consent withdrawal, please see our Privacy Policy

Reviewer #1: No

---

## [Editor Report · Acceptance letter]

PONE-D-25-45283R1

PLOS One

Dear Dr. Botchway,

I'm pleased to inform you that your manuscript has been deemed suitable for publication in PLOS One. Congratulations! Your manuscript is now being handed over to our production team.

Kind regards,

on behalf of

Dr. Burak Tasci

Academic Editor

PLOS One